# Assessing The Importance Of Colours For CNNs In Object Recognition

**Aditya Singh, Alessandro Bay, Andrea Mirabile**
Zebra AI, Zebra Technologies
London, United Kingdom
`{firstname.lastname}@zebra.com`

## Abstract

Humans rely heavily on shapes as a primary cue for object recognition. As secondary cues, colours and textures are also beneficial in this regard. Convolutional neural networks (CNNs), an imitation of biological neural networks, have been shown to exhibit conflicting properties. Some studies indicate that CNNs are biased towards textures whereas, another set of studies suggests shape bias for a classification task. However, they do not discuss the role of colours, implying its possible humble role in the task of object recognition. In this paper, we empirically investigate the importance of colours in object recognition for CNNs. We are able to demonstrate that CNNs often rely heavily on colour information while making a prediction. Our results show that the degree of dependency on colours tend to vary from one dataset to another. Moreover, networks tend to rely more on colours if trained from scratch. Pre-training can allow the model to be less colour dependent. To facilitate these findings, we follow the framework often deployed in understanding role of colours in object recognition for humans. We evaluate a model trained with congruent images (images in original colours eg. red strawberries) on congruent, greyscale, and incongruent images (images in unnatural colours eg. blue strawberries). We measure and analyse network's predictive performance (top-1 accuracy) under these different stylisations. We utilise standard datasets of supervised image classification and fine-grained image classification in our experiments.

## 1 Introduction

Colours play a vital role in our day to day life. We utilise colours for visual identification [1], search [2], gaze guidance in natural scenes [3] etc. As an example, importance of colour can be understood whilst identifying ripe fruits in a background of foliage [1]. Initially, it was widely believed that only shapes and not colours influence object recognition in humans [4, 5]. However, many studies [6, 7] indicate that colours do assist object recognition in humans. The findings by Tanaka and Presnell [6] show that colours facilitate recognition of high colour diagnostic objects (natural objects like fruits) but have little effect on low colour diagnostic objects (man-made objects like airplanes). Their experiments were based on a variation of Stroop effect [8]. Human participants were asked to name objects in different colour schemes. They observed that naming of objects with congruent colours was much faster than naming incongruently coloured objects. Greyscale images served as a neutral medium and the response time for them were in between congruent and incongruent images. In a similar study conducted by Hagen et al. [9] for investigating the role of colour in expert object recognition, similar findings were reported. Neural networks are models of machine learning designed to mimic the neurological working of a human brain [12]. Today, they are employed in many different fields solving numerous tasks [13–15]. Considering the widespread adoption and blackbox nature of neural networks, considerable studies have also been performed to understand their inner working

2nd Workshop on Shared Visual Representations in Human and Machine Intelligence (SVRHM), NeurIPS 2020.

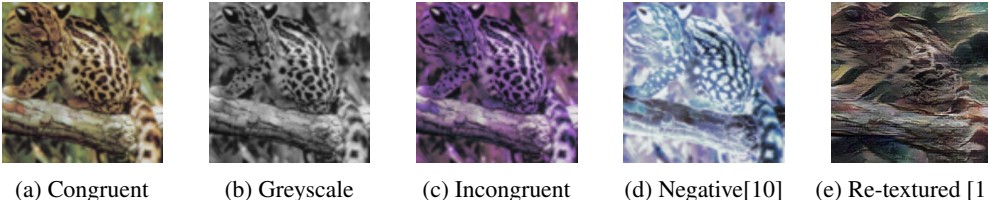

| (a) Congruent | (b) Greyscale | (c) Incongruent | (d) Negative[10] | (e) Re-textured [11] |

Figure 1: The texture and shape information is intact in original, greyscale and incongruent images. Additionally, negative images have a larger drift in distribution than incongruent images when measured against congruent images.

[16–19]. Zeiler and Fergus [16] illustrated the hierarchical nature of learnt features. Engilberge et al. [20] investigated the colour sensitivity of neural networks and observed that at the shallow end the neurons are more sensitive to colour information in an image. A number of existing studies highlight the nature of representations learnt by the network however, do so with conflicting results. One set of results shows that neural networks rely predominantly on shapes [21–25]. On the other hand, many more approaches oppose the theory of shape bias in CNNs [26, 10, 27–29]. Rather than primarily relying on shapes, neural network's predictions are guided by the texture information of an image. Texture is referred to as "a function of the spatial variation in pixel intensities" [30, 31]. Gatys et al. [32] showed that a linear classifier based on texture representations of a neural network performs comparable to the original model. Similarly, Geirhos et al. [11] demonstrated that ImageNet [33] trained model is biased towards texture.

The objective of our paper is to help bridging the gap between human perception and artificial intelligence, providing empirical experiments based on classical neuroscience framework to exhaustively investigate the dependency of CNNs on colour information. We believe the majority of existing approaches do highlight representation bias of CNNs but fail to address the role of colours. Bahng et al. [34] tend to the issue of colour bias in their experiments, however, do so with the motivation to learn unbiased representations. Moreover, by either focusing on small number of test images [21, 29] or a single dataset [11, 26] we are unable to observe a bigger picture.

We believe a fair approach of highlighting the importance of colour is by utilising the framework as used by the psychophysical experiments of [1, 6, 9, 35]. We can easily observe the relevance of colours by comparing performances on congruent and greyscale images. Additionally, by comparing the performances on greyscale and incongruent images we will be able to observe the effect of incorrect colour information (see Figure 4 for sample images).

In this paper, we evaluate the importance of colours for numerous datasets under 2 settings:

1. **Local Information** (Section 4.1): Evaluating the importance of colours when a CNN can only tend to small patches in a global shape agnostic manner.
2. **Global Information** (Section 4.2): For this setting, no such restriction is applied on the network and corresponds to standard approach of training a CNN.

Apart from these 2 modes of experimentation, we also evaluate a model under 4 different training schemes to replicate a typical training scenario for a classification task. We train a network (i) from scratch, (ii) with fine-tuning on pre-trained weights, (iii) with colour based augmentations (random hue, saturation, brightness, etc.), and (iv) with incongruent images to reduce colour dependency. We conclude in Section 5.

## 2    Data

In our study we have employed datasets from image classification and fine-grained visual classification. The datasets used are: • CIFAR-100 [36] • STL-10 [37] • Tiny ImageNet [38] • CUB-200-2011 [39] • Oxford-Flowers [40] • Oxford-IIIT Pets [41]. Table 5 in appendix provides some common statistics of the datasets used.

# 3 Method

A dataset $\mathcal{D} = \{(x_i, y_i), i = 1, \ldots, N\}$ is composed of images $x_i \in \mathbb{R}^{C \times H \times W}$ and their corresponding labels $y_i$. $\mathcal{D}^{train}, \mathcal{D}^{test}$ denotes the split of the dataset into train and test sets respectively. We convert $\mathcal{D}$ into different colour schemes as described below:

- **Congruent Images** ($\mathcal{D}_C$): These are the images in original colours. All the subsequent transformations described below are applied to $\mathcal{D}_C$.
- **Greyscale Images** ($\mathcal{D}_G$): The congruent images converted into greyscale(luminance) images, $G = 0.29 * r + 0.587 * g + 0.114 * b$. We copy the single channel greyscale image into 3 channels.
- **Incongruent Images** ($\mathcal{D}_I$): The channels of the congruent images are switched to generate unnaturally coloured images. Formally, the default correspondence for colour channels is as $C[0] = red, C[1] = green, C[2] = blue$. We switch the channels such that the new ordering represents $C[0] = green, C[1] = blue, C[2] = red$. The advantage of it is that firstly, it preserves the texture and secondly, the changes in distribution are more contained than approaches deployed by [10, 11]. For example, the Jensen-Shannon divergence [42] $JS(\mathcal{D}_C^{train}, \mathcal{D}_G^{train}) = 0.06$ and $JS(\mathcal{D}_C^{train}, \mathcal{D}_I^{train}) = 0.1$, whereas same for negative images (as in [10]) and $\mathcal{D}_C^{train}$ is 0.3 for STL-10 dataset. More comparison is provided in appendix A.2.

Figure 4 shows an example of these stylisations. As humans, we learn from our surroundings which we perceive in congruent colours. Since we are following the framework used in identifying role of colours in humans, we train CNNs only on congruent images ($\mathcal{D}_C^{train}$) while evaluating the top-1 accuracy (represented as $Acc$) on the test sets of different stylisations described above.

# 4 Experiments

## 4.1 Access to local information

Baker et al. [29] report that CNNs can represent local shapes however, fail to utilise it in a global context. Similarly, to highlight absence of shape bias in a CNN, Geirhos et al. [11] utilise BagNets[28] to compare model performance under artistic stylisations. BagNets only have access to small patches within an image due to its design and utilises bag-of-features based framework for making a prediction. It does not make use of spatial ordering of the local features hence making it suitable to compare the relevance of colours to local shape and texture information.

We use BagNet-9[1] which has a $9 \times 9$ receptive field over the image and is built upon the Resnet-50 architecture. We report the mean accuracy and standard deviation over 3 runs. The network is trained from scratch and the data augmentations are limited to random horizontal flips, random rotation and random cropping. The details on hyper-parameters to train the network are provided in the supplementary document.

## 4.1.1 Results

Figure 2 lists the relative accuracies w.r.t $\mathcal{D}_C$ for different stylisations($\frac{\mathcal{D}_G}{\mathcal{D}_C}$ and $\frac{\mathcal{D}_I}{\mathcal{D}_C}$). When comparing $Acc(\mathcal{D}_C)$ with corresponding $\mathcal{D}_I$ and $\mathcal{D}_G$, we observe significant drop in performance. We can also see the varying nature of the gaps in performance across the datasets. This shows that the colours do influence a network but in varying degree. Moreover, for STL-10 and Oxofrd-IIIT Pets, if we compare $Acc(\mathcal{D}_C)$ to $Acc(\mathcal{D}_G)$ we observe that the drop in accuracy is there but comparatively less than the other datasets. This suggests that the network also relies on non-colour features (such as local shapes, textures) to make a decision. Additionally, by comparing $Acc(\mathcal{D}_G)$ and $Acc(\mathcal{D}_I)$ we can notice the consistently lower performance for the latter. This suggests that incorrect colour information does indeed harm the prediction accuracy.

These results indicate that even though a CNN can represent local shape [29] or is biased towards texture[11], colours plays an important part at a local setting.

---

[1]Sourced from official github implementation

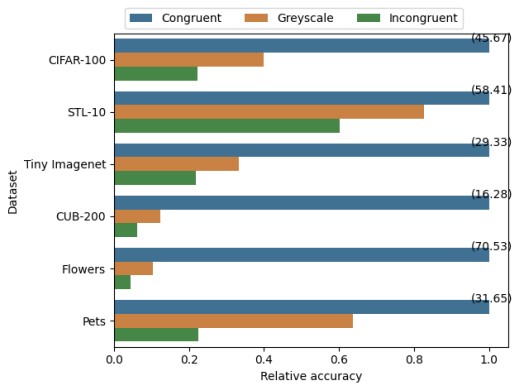

Figure 2: Performance of BagNet-9 on different test stylisations

## 4.2 Access to global information

In the previous experiment, we limited the view of the network to only attend to $9 \times 9$ patches of the image in a global shape agnostic fashion. Here, we use ResNet-18 which has no such restriction on its receptive field. It can thus tend to global shape information in the image.

To assess the importance of colours we train a network in the following 4 ways:

1. **Vanilla training**: Similar to the setting in Section 4.1, we train the network from scratch. The data augmentations used are random rotation, random cropping and random horizontal flips.
2. **Fine-tuning**: In practice, often an ImageNet [33] pre-trained network is fine tuned on the dataset at hand [43]. In this experiment, we follow this protocol keeping the augmentation strategy identical to vanilla training.
3. **Fine-tuning with Augmentations**:
   - **Colour augmentation**: Often colour based augmentations are used in training [44, 45] allowing the network to be colour invariant. All the training settings are identical to fine-tune except for the fact that we also add random colour jitter (hue, saturation, contrast, and brightness) to an image while training.
   - **Channel switch pre-training**: Many of the previous works studying the shape and texture bias have imposed learning limitations by first training the network on stylised data [10, 11] and then fine-tuning on original images. This has been shown to improve predictive performance of the model. Following this approach, we first start with an ImageNet pre-trained model (on congruent images). Then we fine-tune the network following 'colour augmentation' protocol with *randomised channel switching* as an additional data augmentation method. We do this in order to emulate stylised pre-training[11]. After fine-tuning the model, we disable the colour augmentations ('colour jitter' and 'randomised channel switching') and fine-tune the network further only on congruent images.

All the hyper-parameter details are provided in the appendix (see Appendix A.4). We also provide vanilla training results for MobileNet-v2 and DenseNet-121 alongside ResNet-18 and BagNet-9 (see Appendix 4.3).

### 4.2.1 Results & Observations

The results are shown in figure 3. Detailed results are available in the appendix (see appendix A.3). We can draw the following observations from the results.

1. For vanilla and fine-tuned networks, $Acc(\mathcal{D}_C^{test}) > Acc(\mathcal{D}_G^{test}) > Acc(\mathcal{D}_I^{test})$. This trend is similar to what existing studies report for object recognition by humans[9, 35]. However, the difference in CNN accuracies across stylisations are significantly larger. Apart from scoring human participants solely based on accuracy, their response time is also taken into account. For a CNN, there is no variation in the inference time as the architecture remains

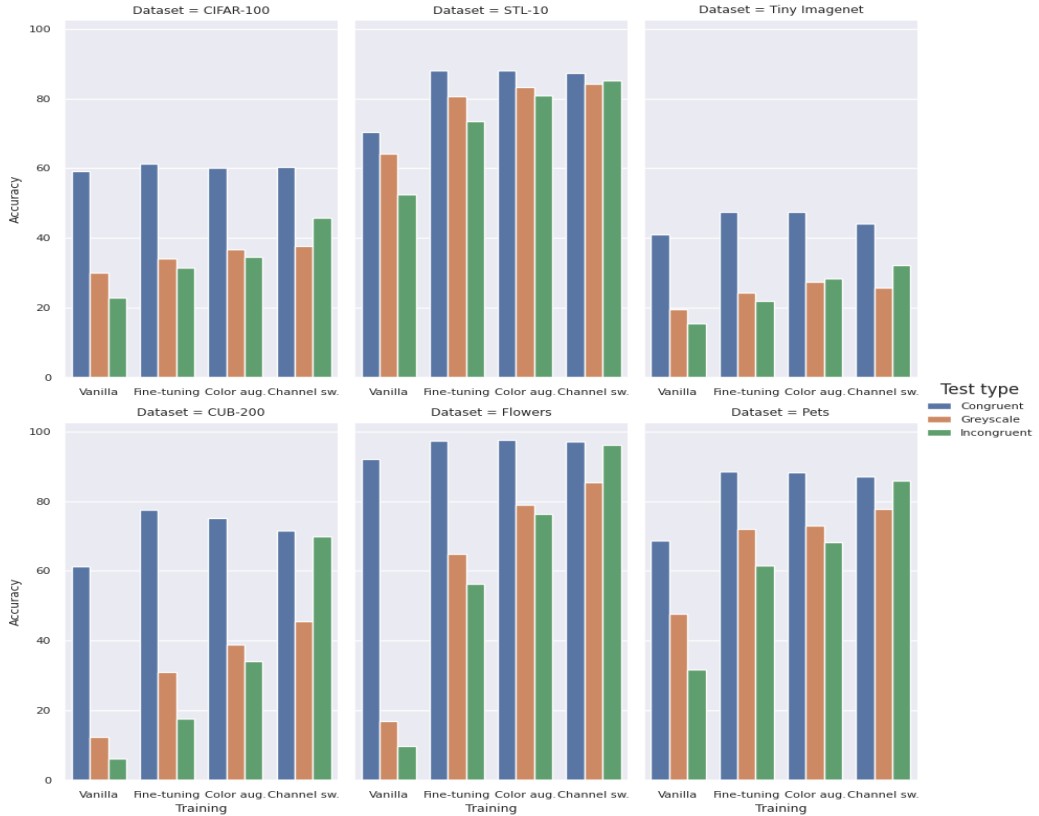

Figure 3: Top-1 $\mathcal{D}^{test}$ accuracies for different training strategies.

constant. However, it can be an interesting extension to understand the differences arising over the predicted estimate. For instance, many approaches utilise predicted value for the winning category as a network's confidence in its prediction [46]. The aim of the study will be then to observe the potential impact of colours on its confidence estimate.

2. Fine-tuning a pre-trained model is widely known to improve the learnt representations of a model and subsequently its accuracy. We observe the additional benefit of fine-tuning which leads to better performance for greyscale and incongruent images indicating lower dependency on colours.

3. Incorporating colour augmentations and channel-switching into training can enforce a model to further rely less on colours. But, it does not improve the network's accuracy for congruent images.

4. The variability for cross-style performance is high across datasets. For example, in the vanilla training setting CUB-200 shows a significantly low performance for greyscale when compared to Oxford-IIIT Pets. We make a similar observation when comparing STL-10 with CIFAR-100. One common property of STL-10 and Pets is that they consist of relatively smaller number of classes (10 and 37 respectively) when compared to CIFAR-100 and CUB-200 (100 and 200 respectively). A direction for future work can be to investigate the relationship between number of categories in the dataset and colour dependency of a CNN. Apart from exploring the dependency over the number of categories we can also investigate if this variation is dependent on categories in the dataset. As humans, we rely more on colours for recognising natural objects than man-made objects [6]. This property is referred to as colour diagnosticity. A similar observation if it exists for CNNs can be worth exploring.

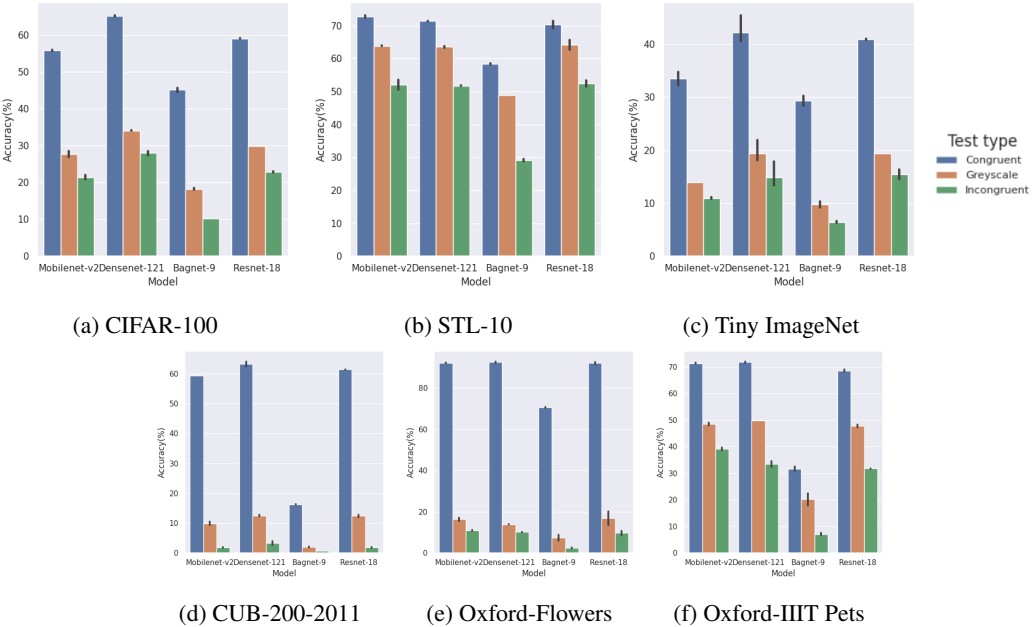

(a) CIFAR-100     (b) STL-10     (c) Tiny ImageNet

(d) CUB-200-2011     (e) Oxford-Flowers     (f) Oxford-IIIT Pets

Figure 4: Vanilla training results for different architectures

### 4.3 Vanilla Performance Across Architectures

In this experiment we include MobileNet-v2 [47] and DenseNet-121 [48] along with ResNet-18 and BagNet-9 to compare their performance across different datasets. This way we can examine if architectural differences play a role in colour bias.

We report the results on different CNN architectures trained under the vanilla setting in Figure 4. The results show that different architectures display similar behaviour for colour importance across datasets. On the congruent images, the networks perform the best whereas the performance is worst for incongruent images. This shows that the underlying the architecture plays a less significant role in driving the bias of a network towards colour. The importance of colour is more dependent on the task at hand.

## 5 Conclusion

We believe ours is the first work to recognise unattributed impact of colours to the shape/texture driven research for understanding bias in CNNs. By adopting the psychophysical experiment for CNNs, we have provided empirical evidence to highlight high impact of colours. We showed that a variety of different CNNs show high colour dependency for the classification task. This dependency appears to be tied to the dataset than the underlying architecture. By default, the networks are highly colour dependent and this dependency can be reduced by utilizing pre-trained weights and employing various augmentations in training as showed in our work.

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

# A   Appendix

## A.1   Data

| Category | Dataset | #Classes | $|\mathcal{D}^{train}|$ | $|\mathcal{D}^{test}|$ | Image size |
|---|---|---|---|---|---|
| Image classification | CIFAR-100[36] | 100 | $50,000$ | $10,000$ | $32 \times 32$ |
| | STL-10[37] | 10 | $5,000$ | $8,000$ | $96 \times 96$ |
| | Tiny ImageNet[38] | 200 | $100,000$ | $5,000$ | $64 \times 64$ |
| Fine-Grained Visual Classification | CUB-200[39] | 200 | $5,994$ | $5,794$ | $224 \times 224$ |
| | Oxford-Flowers[40] [2] | 102 | $6,149$ | $2,040$ | $224 \times 224$ |
| | Oxford-IIIT Pets[41] | 37 | $3,680$ | $3,669$ | $224 \times 224$ |

Table 1: Dataset statistics

## A.2   Jensen-Shannon Measure

To reiterate the notations used, a dataset $\mathcal{D} = \{(x_i, y_i), i = 1, \ldots, N\}$ is composed of images $x_i \in \mathbb{R}^{C \times H \times W}$ and their corresponding labels $y_i$. $\mathcal{D}^{train}$, $\mathcal{D}^{test}$ denotes the split of the dataset into train and test sets respectively. Jensen Shannon divergence between two stylisations of the same dataset is defined as:

$$JS(\mathcal{D}_1^{train}, \mathcal{D}_2^{train}) = \frac{1}{|\mathcal{D}^{train}|} \sum_{i \in \mathcal{D}^{train}} \frac{1}{|C|} \sum_{j \in C} JS(T_1(x_i)[j], T_2(x_i)[j])$$

where, $T_i$ is the transformation corresponding to $\mathcal{D}_i$ and indexing $[j]$ returns the normalised intensity histogram for the $j^{th}$ channel.

The corresponding results in Table 2 show that switching channels is a gentler transformation than composing negatives in preserving original shape and texture. Greyscale is consistently with the least amount of JS divergence and can suggest as to why consistently $\text{Acc}(\mathcal{D}_G) > \text{Acc}(\mathcal{D}_I)$.

Table 2: JS measure between stylisations

| Datasets | $JS(\mathcal{D}_C^{train}, \mathcal{D}_G^{train})$ | $JS(\mathcal{D}_C^{train}, \mathcal{D}_I^{train})$ | $JS(\mathcal{D}_C^{train}, \mathcal{D}_{Neg}^{train})$ |
|---|---|---|---|
| CIFAR-100 | 0.07 | 0.12 | 0.33 |
| Tiny ImageNet | 0.06 | 0.1 | 0.29 |
| STL10 | 0.06 | 0.1 | 0.3 |
| CUB-200 | 0.1 | 0.15 | 0.34 |
| Oxford-Flowers | 0.09 | 0.13 | 0.32 |
| Oxford-IIIT Pets | 0.04 | 0.07 | 0.27 |

## A.3   Detailed Test results

## A.4   Training details

We utilised Pytorch framework for all of our experiments. We list the detailed hyper-parameters below. Missing key-values in table $i$ can be found via. recursive search in table $i - 1$. All the fine-grained datasets utilise similar training hyper-parameters and hence we have provided only the details for CUB-200.

Table 3:
Test accuracies(in %) for different training strategies on classification datasets.

| Training approach | Datasets | Acc($\mathcal{D}_C$) | Acc($\mathcal{D}_G$) | Acc($\mathcal{D}_I$) |
|---|---|---|---|---|
| Vanilla | CIFAR-100 | $59.10 \pm 0.11$ | $29.90 \pm 0.03$ | $22.86 \pm 0.16$ |
| | STL-10 | $70.30 \pm 1.67$ | $64.13 \pm 2.13$ | $52.47 \pm 1.29$ |
| | STL-10 | $40.88 \pm 0.10$ | $19.40 \pm 0.02$ | $15.47 \pm 1.31$ |
| Fine-tuning | CIFAR-100 | $\mathbf{61.39} \pm 0.12$ | $34.13 \pm 0.16$ | $31.53 \pm 0.55$ |
| | STL-10 | $\mathbf{88.10} \pm 0.30$ | $80.59 \pm 0.39$ | $73.45 \pm 0.69$ |
| | STL-10 | $\mathbf{47.52} \pm 0.09$ | $24.38 \pm 0.21$ | $21.96 \pm 0.19$ |
| Colour augmentation | CIFAR-100 | $60.06 \pm 0.29$ | $36.68 \pm 0.82$ | $34.67 \pm 0.24$ |
| | STL-10 | $88.04 \pm 0.23$ | $83.29 \pm 0.46$ | $80.88 \pm 0.06$ |
| | STL-10 | $47.42 \pm 0.39$ | $\mathbf{27.41} \pm 1.10$ | $28.27 \pm 0.09$ |
| Channel switch pre-training | CIFAR-100 | $60.24 \pm 0.22$ | $\mathbf{37.76} \pm 0.26$ | $\mathbf{45.84} \pm 0.44$ |
| | STL-10 | $87.20 \pm 0.45$ | $\mathbf{84.01} \pm 0.19$ | $\mathbf{85.23} \pm 0.21$ |
| | Tiny Imagenet | $44.20 \pm 0.16$ | $25.75 \pm 0.46$ | $\mathbf{32.11} \pm 0.02$ |

Table 4:
Test accuracies(in %) for different training strategies on fine-grained datasets.

| Training approach | Datasets | Acc($\mathcal{D}_C$) | Acc($\mathcal{D}_G$) | Acc($\mathcal{D}_I$) |
|---|---|---|---|---|
| Vanilla | CUB | $61.39 \pm 0.03$ | $12.40 \pm 0.58$ | $6.03 \pm 0.23$ |
| | Oxford-Flowers | $92.03 \pm 0.72$ | $16.88 \pm 4.81$ | $9.68 \pm 1.42$ |
| | OP | $68.61 \pm 0.63$ | $47.75 \pm 0.80$ | $31.77 \pm 0.15$ |
| Fine-tuning | CUB | $\mathbf{77.39} \pm 0.40$ | $30.99 \pm 0.36$ | $17.62 \pm 2.30$ |
| | Oxford-Flowers | $97.37 \pm 0.03$ | $64.77 \pm 1.42$ | $56.27 \pm 3.32$ |
| | OP | $\mathbf{88.51} \pm 0.86$ | $72.06 \pm 1.15$ | $61.59 \pm 2.11$ |
| Colour augmentation | CUB | $75.07 \pm 0.09$ | $38.85 \pm 1.33$ | $34.09 \pm 0.15$ |
| | Oxford-Flowers | $\mathbf{97.50} \pm 0.14$ | $78.82 \pm 0.67$ | $76.37 \pm 1.82$ |
| | Pets | $88.15 \pm 0.32$ | $72.86 \pm 0.05$ | $68.22 \pm 1.04$ |
| Channel switch pre-training | CUB | $71.5 \pm 0.67$ | $\mathbf{45.41} \pm 0.32$ | $\mathbf{69.93} \pm 0.57$ |
| | Oxford-Flowers | $97.15 \pm 0.06$ | $\mathbf{85.44} \pm 1.24$ | $\mathbf{96.17} \pm 0.21$ |
| | OP | $87.16 \pm 1.04$ | $\mathbf{77.65} \pm 1.29$ | $\mathbf{85.77} \pm 1.11$ |

Table 5: Training details for CIFAR-100

| Approach | Key | Value |
|---|---|---|
| Common | Models | Bagnet-9, Resnet-18, Densenet-121, Mobilenet-v2 |
| | Image size | $32 \times 32$ |
| | Train aug. | Random(rotation, horizontal flip), standardisation |
| | Test aug. | Standardisation |
| | Batch size | 128 |
| | Optimiser | SGD |
| | LR decay rate | 0.5 |
| Vanilla | Epochs | 200 |
| | LR | 0.1 |
| | Train aug. | Common |
| | LR decay epochs | [50, 100, 150] |
| Fine-tuning | Pre-trained weights | ImageNet |
| | LR | 0.01 |
| | Epochs | 30 |
| | LR decay epochs | [15] |
| Colour augmentation | Pre-trained weights | ImageNet |
| | Train aug. | Common + Random colour jitters |
| | LR | 0.01 |
| | Epochs | 30 |
| | LR decay epochs | [15] |
| Incongruent Training | Pre-trained weights | ImageNet |
| | Train aug. | Common + Random (colour jitters, channel switching) |
| | Finetuned with | Random(rotation, horizontal flip) |
| | LR | 0.01 |
| | Epochs | 30, 30 |
| | LR decay epochs | [15], [15] |

Table 6: Training details for STL-10

| Approach | Element | Value |
|---|---|---|
| Common | Image size | $96 \times 96$ |
| | Batch size | 64 |
| Vanilla | Epochs | 200 |
| | LR | 0.1 |
| | LR decay epochs | [40, 80, 120, 160] |
| Fine-tuning | Epochs | 50 |
| | LR | 0.01 |
| | LR decay epochs | [15, 30, 45] |
| Colour augmentation | Epochs | 50 |
| | LR | 0.01 |
| | LR decay epochs | [15, 30, 45] |
| Incongruent Training | Epochs | 50, 50 |
| | LR | 0.01 |
| | LR decay epochs | [15, 30, 45], [15, 30, 45] |

Table 7: Training details for Tiny ImageNet

| Approach | Element | Value |
|---|---|---|
| Common | Image size | $64 \times 64$ |
| | Batch size | 128 |
| Vanilla | Epochs | 150 |
| | LR | 0.1 |
| | LR decay epochs | $[30, 60, 90, 120]$ |
| Fine-tuning | Epochs | 50 |
| | LR | 0.01 |
| | LR decay epochs | $[15, 30, 45]$ |
| Colour augmentation | Epochs | 50 |
| | LR | 0.01 |
| | LR decay epochs | $[15, 30, 45]$ |
| Incongruent Training | Epochs | 50, 50 |
| | LR | 0.01 |
| | LR decay epochs | $[15, 30, 45], [15, 30, 45]$ |

Table 8: Training details for CUB-200

| Approach | Key | Value |
|---|---|---|
| Common | Image size | $224 \times 224$ |
| | Train aug. | Random(rotation, horizontal flip, crop), standardisation |
| | Test aug. | center crop(224), standardisation |
| | Batch size | 32 |
| | Optimiser | SGD |
| | LR decay rate | 0.5 |
| Vanilla | Epochs | 200 |
| | LR | 0.1 |
| | Train aug. | Common |
| | LR decay epochs | $[50, 100, 150]$ |
| Fine-tuning | Pre-trained weights | ImageNet |
| | LR | 0.01 |
| | Train aug. | Common |
| | Epochs | 40 |
| | LR decay epochs | $[15, 30]$ |
| Colour augmentation | Pre-trained weights | ImageNet |
| | LR | 0.01 |
| | Train aug. | Common + Random colour jitters |
| | Epochs | 40 |
| | LR decay epochs | $[15, 30]$ |
| Incongruent Training | Pre-trained weights | ImageNet |
| | Train aug. | Common + Random (colour jitters, channel switching) |
| | Finetuned with | Common aug. |
| | LR | 0.01 |
| | Epochs | 40, 40 |
| | LR decay epochs | $[15, 30], [15, 30]$ |

