# OpenReview forum: "Assessing The Importance Of Colours For CNNs In Object Recognition"
_NeurIPS.cc/2020/Workshop/SVRHM — SVRHM@NeurIPS Poster_

### Official Review · AnonReviewer3 · 2020-10-27
**Do humans and deep networks rely on colour information similarly?**

**Rating:** 7
**Confidence:** 5

**Review:**

I find this line of investigation to study the impact of colour and its representation in deep networks very interesting and important for human- and machine-vision community. It would be nice to read more about the shared representation between humans and machines in the conclusion/discussion section linking (more strongly) the findings of this article to human colour vision.

Pro: This paper is clearly written and easy to read.

Con: The number of tasks and studied datasets is limited.


Major comments:
- "Neural networks are models of machine learning designed to mimic the neurological working of a human brain [12]."
I agree that there are many similarities between the biological and artificial neural networks, however, I'm not sure to what extent the networks were designed to mimic the human brain.
For instance, very relevant to this study, the human visual system consists of three processing channels (luminance, red-green and yellow-blue) that coexist in parallel at least to a large part of V1. Contrary to this, the chromatic information is often (almost always) collapsed in the first convolutional layer.
- Authors might be interested in the following articles:
 + Rafegas, I. and Vanrell, M., 2018. Color encoding in biologically-inspired convolutional neural networks. Vision Research, 151, pp.7-17.
Flachot, A. and Gegenfurtner, K.R., 2018. Processing of chromatic information in a deep convolutional neural network. JOSA A, 35(4), pp.B334-B346.
These articles show that colour opponency emerges in deep networks.
 + Akbarinia, Arash, and Raquel Gil-Rodríguez. "Deciphering image contrast in object classification deep networks." Vision Research 173 (2020): 61-76.
This article shows edges (contours of shapes) are of importance to object classification networks.
- It would be nice to analyse the difference between C100, TIN and S10 to better understand why S10 is less dependent on colour:
 + Quantitative analysis, for instance, the distribution of colour in those datasets.
 + Visualising a few images from each dataset to facilitate their comprehension for readers.

Minor comments:
- It would be nice to have the reference for used datasets in Table 1 for reproduction purposes.
- I think the first paragraph of Section 4.1 is more appropriate to be placed in the introduction when the concept of shape-texture is compared.
- What is VA in P4-L113?
- It would be nice to have consistency in the order of datasets in Tables 1-4.

---

### Official Review · AnonReviewer1 · 2020-10-29
**Assessing The Importance Of Colours For CNNs In Object Recognition**

**Rating:** 6
**Confidence:** 4

**Review:**

## Summary

Motivated by the literature on the effect of color on object recognition in humans, as well as by recent experiments probing shape and texture in CNNs, this paper considers the role of color information in CNN classification performance. The authors train 2 architectures (BagNet-9 and ResNet-18) on 6 datasets (3 standard - e.g. Tiny ImageNet, 3 fine-grained - e.g. Oxford-Flowers) and compare performance on color-congruent, grayscale, and color-incongruent versions of test images. They additionally consider models trained from scratch versus fine-tuned from an ImageNet-pretrained model (with and without data augmentations that act on color). They find that, like people, across architectures and datasets, models trained from scratch or finetuned without color augmentation perform better on color-congruent images, with accuracy dropping off for grayscale, and still more so, for color-incongruent versions of the test sets. In general, finetuned models perform better on all test sets than do models trained from scratch. Color-jitter augmentation generally increases performance on grayscale and color-incongruent images relative to the un-augmented fine-tuned baseline. Data augmentation consisting of random channel switching generally further increases performance on these images.

## Pros
The general research question is well motivated and connects with existing work in both psychophysics/visual neuroscience and recent investigations into CNN behavior. This paper contains a large number of experiments with a range of datasets and with two model architectures. The authors consider the effects of two types of data augmentation that act on color.

## Cons
### Presentation of data
In general, it was effortful to extract the main points from the tables. A couple things that would help:
* There is space to write out the full names of the datasets (e.g. "Oxford Flowers" rather than "OF"), and this will be useful for a wider audience less familiar with these datasets. Likewise, spell out "D_{congruent}" instead of "D_C", etc.
* I encourage the authors to consider replacing the tables with plots, e.g. bar plots with error bars, where possible. In this case, please include lines to indicate chance performance for each dataset.
* Where impractical to present as plots, please guide the reader by describing in the caption what is bolded.

### Methods (4.2)
Several details were unclear, namely:
* Whether data augmentation was applied during the ImageNet pretraining, during finetuning, or both
* Whether channel switch pre-training was done in combination with the "vanilla" or "fine-tuning" training.
* This statement sounds as though it applies to both (3 - colour augmentation) and (4 - channel switch pre-training): "we proceed to fine-tune the network further without 'randomised channel switching' and 'colour jitter'". How long was the network fine-tuned without augmentation, and what was the reason for this choice?
* Assuming my understanding is correct, I recommend a reorganization to make clear that the augmentation was applied in combination with fine-tuning: e.g.
		1. Vanilla training: (…)
		2. Fine-tuning an ImageNet-pretrained model
			i. Without augmentation: (…)
			ii. With augmentation: (…shared details…)
				1) Color jitter (…)
				2) Channel switch (…)

### Color-incongruent images
The authors created color-incongruent images by reversing the channel order from [RGB] --> [GBR], yielding images like the sample presented in Figure 1c. It is noticeable that, rather than changing just object color, this transformation changes the color of the entire image, foreground and background alike. This is a divergence from the stimulus sets used to test congruent/incongruent processing in people (e.g. for a recent example: Teichmann et al. 2020, https://www.jneurosci.org/content/40/35/6779). While there is no doubt that the manipulation acts on color information, it would be worth elaborating on why this particular transformation was chosen, and potential drawbacks (if any).

### Claims about shape
Throughout, the authors make claims about whether models are using shape information, e.g. "colour augmentations or channel-switching into training can enforce a model to perform based on shapes" and "fine-tuning…helps to focus more on shapes, evident from improved performance on greyscale and incongruent images." The fact that performance improves on these test sets is not on its own evidence that a model is using shape specifically given that other features, e.g., texture and image background, are also preserved across color transformations. In the conclusion, the authors state, "we showed that the common practice of fine-tuning allows for less colour dependent features and improved performance on shape dependent tasks". Again, there is a strong assumption here that the datasets in question require shape and cannot be done using other features, like texture or background context.

### Take-aways
 Models generally perform worse on out-of-distribution test images (here, grayscale and color-incongruent images) than on in-distribution ones (here, color-congruent images). It would be useful for the authors to sharpen their take-away points to highlight the specific contributions of the current study. The authors note briefly but do not elaborate on differences between model architectures and datasets. Is there anything interesting to be gleaned here? What do the differences between datasets suggest about the potential relevance of color for different classification tasks? The authors remark, "It encourages to think if colour diagnosticity is a phenomenon that we can expect a deep neural network to exhibit", but do not elaborate.

### Related work
* Kubilius et al. (2016) compared CNN classifications of color versus grayscale versus silhouette Snodgrass & Vanderwart images (drawings). They found that CNNs performed better on color than grayscale (15% drop), and better on grayscale than silhouettes (30% drop). Geirhos et al. 2019 found that an ImageNet-trained ResNet-50 and a Stylized ImageNet-trained ResNet-50 performed roughly comparably on natural images and grayscaled versions of those images. It would be useful to briefly connect with these (differing) results.
* There have been several prior papers examining (intermediate) color representations in ImageNet-trained CNNs, and which may be worth citing (the approach differs from the one taken here):

Flachot, A., & Gegenfurtner, K. R. (2018). Processing of chromatic information in a deep convolutional neural network. JOSA A, 35(4), B334-B346.

Rafegas, I., Vanrell, M., Alexandre, L. A., & Arias, G. (2020). Understanding trained CNNs by indexing neuron selectivity. Pattern Recognition Letters, 136, 318-325.

Rafegas, I., & Vanrell, M. (2017). Color representation in CNNs: parallelisms with biological vision. In Proceedings of the IEEE International Conference on Computer Vision Workshops (pp. 2697-2705).

## Overall evaluation
Overall, the quality, clarity, originality and significance of this work is marginal. However, on balance, this data is useful for the community, and so I recommend acceptance with the hope that the overall presentation and discussion will be improved.

---

### Official Review · AnonReviewer2 · 2020-10-29
**Interesting topic, neat experiments and well written**

**Rating:** 8
**Confidence:** 5

**Review:**

## Summary of the paper

This paper explores empirically the role of colour in artificial neural networks trained for image object categorisation. The main aspect of the experiments is the comparison in performance of trained models when the test images are transformed into greyscale, the colours are incongruent (through channel switching), and the colours are congruent with the training images. The main conclusion is that performance drops considerably if colour information is missing or is incongruent, though large differences exist across data sets. Second, the authors concluded from their experiments that pre-training on ImageNet significantly improves the performance on congruent images (unsurprisingly), but also on the colour-altered test images. Third, training with colour-dependent transformations reduced the performance, suggesting that colour is important for accurate object recognition.

## Summary of merits and concerns

### Merits

+ The paper touches on a relevant topic, the role of colour in machine vision, which has not been sufficiently explored in the literature.
+ It presents the results from a set of reasonable experiments which shed light on the dependence of neural networks on colour information, as well as the effect on it of common techniques such as pre-training and data augmentation.
+ Despite some aspects that may be improved (see below), the paper is generally well written: the introduction nicely motivates the topic of the paper and reviews relevant previous work from both the machine learning and the cognitive science literature. The experimental setup is clear, largely reproducible with the help of the supplementary material, the main results are discussed in the text and the conclusion, though brief, provides a good summary of the paper.

### Concerns

- It is hard to digest and navigate the results offered in the tables. As I discuss below, I believe that presenting the results graphically and offering more direct, relevant comparisons would make the paper stronger.
- Although transparent and reasonable, the experimental setup is limited regarding the breadth of neural networks. Since only one architecture is evaluated, the question of whether the results apply only to ResNet or more generally remains open.
- No statistical analysis is provided to quantify the conclusions and support the claims.

## Evaluation and justification

I have a generally positive impression of this paper as it addresses a topic, the role of colour in artificial neural networks trained for image object recognition, which as not received much attention by the machine learning community, being an important topic in visual perception and cognitive neuroscience. The experiments presented in the article are meaningful and allow do draw some conclusions, though preliminary, about how altering the colour information of images impacts the object recognition performance. Furthermore, the paper is generally well written. I still have some concerns on which I elaborate below with the intention of providing constructive feedback, but these are outweighed by the contributions. Hence, I recommend the acceptance of this paper for its presentation at SVRHM.

### Broader experiments desirable

First of all, I very positively appreciate that the authors carried out experiments with up to 6 different data sets. That definitely offers consistency with respect to data sets, plus it allows to observe relevant differences too. However, I think that the generality of the conclusions would be much better if experiments on additional architectures were carried out. The current version of the paper presents results only on a ResNet architecture, and a subset of experiments on BagNet. In my opinion, extending the experimental setup in this direction would be desirable for potential future work or versions of the paper.

### Visualisation of results

I see substantial room for improvement regarding the way the results of the experiments are presented. I am aware that it is a common practice in machine learning to report big tables full of numbers to present the performance of neural networks or other kind of the results. However, this is, in my humble opinion, highly suboptimal. Making sense of these big tables requires considerable cognitive load and it hinders the interpretability. A preferable alternative, again in my opinion, is to display the results graphically. Take as an example the results in [1], where colour bar plots are used, although an even better alternative is to represent the performance of each model with a dot or a box plot if there were multiple data points.

This has a number of advantages: 1) High-level conclusions can be made easily and quickly: for example, we would see immediately that the accuracy on greyscale images (dots in one colour, for instance) is lower than on the congruent images (in another colour), without having to read, interpret and compare multiple pairs of numbers 2) It allows to pay attention to the relevant comparisons, which can be highlighted with colours, shades, markers, etc.

Furthermore, I also suggest to present the results of the models in relative terms with respect to a baseline, instead of the absolute accuracy. This is particularly suitable for this study because most results are compare to a baseline and the fact that the results correspond to multiple data sets, with different baselines, makes the comparison across data sets nearly impossible. Let me take the results in Table 3 to illustrate what I mean. It seems that the baseline should be the results of the _Vanilla_ model on congruent images. The rest of the results could be presented as accuracy / baseline * 100, which would indicate the fraction of the baseline accuracy (or improvement) achieved by each model---this is also done in [1], and [2] is a good reference on how to report experimental results. Not only does this make the results more easily interpretable, but also allows comparisons across data sets and, in turn, statistical analysis, which the final aspect I miss in this paper.

## Questions

Below I list some questions that I had while reading the paper. I would be interested in the answers to satisfy my own curiosity, but mainly they may be considered as constructive feedback:

- In the fine-tuning strategy, are all weights updated or only the top layers, another strategy?
- A relevant, general question that permeates the topic of this paper is: Is it desirable that models are invariant to colour channel switches, and greyscale transformations? It would be interesting to discuss more in depth this question.

## Minor comments and potential typos identified

In the following list I mention some potential typos and make some minor comments that may be considered by the authors to potentially improve their article:

- "A number of studies also exist highlighting the nature of representations learnt by the network however, do so with conflicting results.": Is this sentence correctly structured?
- Typo?: "One set of results _show_ that" (Section 1): should read _shows_ if the subject is "set".
- Typo: "One set of results show that, neural networks": extra comma
- Language: First paragraph in page 2: consider using connectors and articles to improve the reading flow, especially the last two sentences.
- Section 2 Data looks a bit odd to me: First, I think it is important to explicitly provide the full name and reference of each data set (CIFAR-100 instead of C100, STL-10 instead of S10, etc.), since not all readers are familiar with them, plus the abbreviations used are not standard. The authors do write the full names of the data sets in the abstract, but in my opinion, the opposite makes more sense to me: in the abstract it may be enough to say "image classification and fine-grained image classification". Second, since the section is so short and the information about the data is actually part of the methodology, the authors may consider integrating Section 2 into Section 3. This said, I do appreciate positively that the authors included a table (Table 1) with the summary of the data sets' characteristics.
- Language: "Since, we are following": extra comma
+ "We report the mean accuracy and standard deviation over 3 runs": Fantastic! Very much appreciated!
- Typo: "colour jitter(hue": space missing
- Consider developing the acronym "FGVC" in Table 4, since it is the only place where it is used in the paper.
- It is confusing that the order in Table 2 is C100-TIN-S10, but in Table 3 is C100-S10-TIN.
- Typo?: "We believe, ours is the first work": extra comma
- Typo: "Through emprical evaluations": "empirical"

## References

[1] Hernández-García, Alex, and Peter König. "Data augmentation instead of explicit regularization." arXiv preprint arXiv:1806.03852, 2018.

[2] Dodge, Jesse, et al. "Show your work: Improved reporting of experimental results." arXiv preprint arXiv:1909.03004, 2019.

---

### Public Comment · ~Aditya_Singh3 · 2020-12-07
**Thank You!**

We'd like to thank the reviewers for their thorough feedback!
We have incorporated majority of the suggestions put forth in reviews and will be uploading the camera-ready version shortly.

See you all soon!

---

### Decision · Program_Chairs · 2020-11-02

Accept (Poster)